

2                              **Warm Winter, Thin Ice?**
3          Julienne Stroeve[1,2], David Schroder[3], Michel Tsamados[1], Daniel Feltham[3]
[1]Centre for Polar Observation and Modelling, Earth Sciences, University College London,
London, UK
[2]National Snow and Ice Data Center, University of Colorado, Boulder, CO, USA
[3]Centre for Polar Observation and Modelling, Department of Meteorology, University of
Reading, Reading, UK
**Abstract**
Winter 2016/2017 saw record warmth over the Arctic Ocean, leading to the least amount of
freezing degree days north of 70°N since at least 1979. The impact of this warmth was
evaluated using model simulations from the Los Alamos sea-ice model (CICE) and CryoSat-
2 thickness estimates from three different data providers. While CICE simulations show a
broad region of anomalously thin ice in April 2017 relative to the 2011-2017 mean, analysis
of three CryoSat-2 products show more limited regions with thin ice and do not always agree
with each other, both in magnitude and direction of thickness anomalies. CICE is further used
to diagnose feedback processes driving the observed anomalies, showing 11-13 cm reduced
thermodynamic ice growth over the Arctic domain used in this study compared to the 2011-
2017 mean, and dynamical contributions of +1 to +4 cm. Finally, CICE model simulations
from 1985-2017 indicate the negative feedback relationship between ice growth and winter
air temperatures may be starting to weaken, showing decreased winter ice growth since 2012
as winter air temperatures have increased and the freeze-up has been further delayed.
**Introduction**
27           It is well known that Arctic air temperatures are rising faster than the global average [e.g.
*Bekryaev et al.*, 2010; *Serreze and Barry*, 2011]. The thinning and shrinking of the summer
sea ice cover have played a role in this amplified warming, which is most prominent during
the autumn and winter months as the heat gained by the ocean mixed layer during ice-free
summer periods is released back to the atmosphere during ice formation [e.g. *Serreze et al.*,
2009; *Screen and Simmonds*, 2010]. However, Arctic amplification has been found in climate
models without changes in the sea ice cover [*Pithan and Mauritsen*, 2014]. Increased latent
energy transport [*Graversen and Burtu*, 2016], the lapse rate feedback [*Pithan and
Mauritsen*, 2014; *Graversen*, 2006] and changes in ocean circulation [*Polyakov et al.*, 2005]
have also contributed. Furthermore, cyclones are effective means of bringing warm and moist
air into the Arctic during winter [e.g. *Boisvert et al.*, 2016].
38           Winter 2015/2016 was previously reported as the warmest Arctic winter recorded since
records began in 1950 [*Cullather et al.*, 2016]. Warming was Arctic-wide, with temperature
anomalies reaching +5°C [*Overland and Wang*, 2016] and temperatures near the North Pole
hitting 0°C [*Boisvert et al.*, 2016]. Part of the unusual warming was linked to a strong cyclone
that entered the Arctic in December 2015 [*Boisvert et al.*, 2016], resulting in reduced
thermodynamic ice growth and thinning within the Kara and Barents seas [*Ricker et al.*,
2017; *Boisvert et al.*, 2016]. This was one of several cyclones to enter the Arctic that winter
as a result of a split tropospheric vortex that brought warm and moist air from the Atlantic
Ocean towards the pole [*Overland and Wang*, 2016]. Winter 2016/2017 once again saw
temperatures near the North Pole reach 0°C in December 2016 and February 2017 [*Graham
et al.*, 2017]. These warming events were similarly associated with large storms entering the
Arctic [*Cohen et al.*, 2017]. It has been suggested that the recent warm winters represent a





trend towards increased duration and intensity of winter warming events within the central
Arctic [*Graham et al.*, 2017].
In general, warm winters, combined with increased ocean mixed layer temperatures from
summer sea ice loss, delay freeze-up, impacting the length of the ice growth season and the
period for snow accumulation on the sea ice. *Stroeve et al.* [2014] previously evaluated
changes in the melt onset and freeze-up, showing large delays in freeze-up within the
Chukchi, East Siberian, Laptev and Barents seas, with delays increasing on the order of +10
days per decade. Later freeze-up has a non-trivial influence on basin-wide sea ice thickness:
ice grows thermodynamically faster for thin ice than for thick ice [*Bitz and Roe*, 2004]. More
subtle effects involving the timing of ice growth relative to major snow precipitation events
in fall have been shown to also control the growth rate of sea ice thickness; ice grows faster
for a thinner snow pack [*Merkouriadi et al.*, 2017]. Nevertheless, the maximum winter sea
ice extent in 2017 set a new record low for the 3$^{rd}$ year in a row. Have the recent warm
winters played a role in these record low winter maxima by reducing winter ice formation?
*Ricker et al.* [2017a] previously evaluated the impact of the 2015/2016 warm winter on
ice growth using sea ice thickness derived from blending CryoSat-2 (CS2) radar altimetry
with those from Soil Moisture and Ocean Salinity (SMOS) radiometry [*Ricker et al.*, 2017b].
They found anomalous freezing degree days (FDDs) between November 2015 and March
2016 within the Barents Sea of 1000 degree days coincided with a thinning of approximately
10 cm in March compared to the 6-year mean. While near-surface air temperatures largely
control thermodynamic ice growth, other processes also impact ice growth, including ocean
circulation, sensible and latent heat exchanges. Furthermore, winter ice thickness is not only
a result of thermodynamic ice growth, but rather the combined effects of thermodynamic and
dynamic processes. A thinner ice cover is more prone to ridging and rafting, as well as ice
divergence, leading to new ice formation within leads/cracks within the ice pack. This
however was not evaluated by *Ricker et al.* [2017a].
In this study we evaluate the impact of the 2016/2017 anomalously warm winter on
Arctic sea ice thickness using the Los Alamos sea-ice model (CICE) [*Hunke et al.*, 2015] and
satellite-derived CS2 thickness data from three different sources: Centre for Polar
Observation and Modeling (CPOM) [*Tilling et al.*, 2017], Alfred Wegener Institute (AWI)
[*Hendricks et al.*, 2016], and NASA [*Kurtz and Harbeck*, 2017]. CICE is initialized with
CPOM CS2 sub-grid scale ice thickness distribution (ITD) fields in November and run
forward with NCEP Reanalysis-2 (NCEP2) atmospheric reanalysis data [*Kanamitsu et al.*,
2002, updated 2017]. The model run is subsequently compared over the winter growth season
to CS2 thickness from the three different data providers and contributions of thermodynamics
vs. dynamics to the thickness anomalies are evaluated. While the focus is on the 2016/2017
ice growth season, a secondary aim is to compare existing CS2 products to inform the
community on uncertainties in these estimates and inform on model limitations. Thus, results
are also presented for other years during the CS2 time-period for comparison. To our
knowledge, this is the first study to compare different CS2 data products over the lifetime of
the mission.
**Methods**
*Ice Thickness Distribution (ITD) from Cryosat-2*
The CryoSat-2 radar altimetry mission was launched April 2010, providing estimates of
ice thickness during the ice growth season. CS2 provides freeboard estimates, or the height of
the ice surface above the local sea surface, which when combined with information on snow
depth, snow density and ice density can be converted to ice thickness assuming hydrostatic
equilibrium [e.g. *Laxon et al.*, 2013]. Here we evaluate ice thickness fields provided by three
different data providers in order to assess robustness of the observed thickness anomalies.



Thickness is retrieved from ice freeboard by processing CS2 Level 1B data, with a footprint
of 300m by 1700m, and assuming snow density and snow depth from the *Warren et al.*
[1999] climatology (hereafter *W99*), modified for the distribution of multiyear versus first-
year ice (i.e. snow depth is halved over first-year ice) [see *Laxon et al.*, 2013 and *Tilling et*
*al.*, 2017 for data processing details].
While the three data providers rely on *W99* for snow depth and density, each institution
processes the radar returns differently. In general, the range to the main scattering horizon of
the radar return is obtained using a retracker algorithm. This can be based on a threshold [e.g
*Laxon et al.*, 2013; *Ricker et al.*, 2014; *Hendricks et al.*, 2016], or a physical retracker [*Kurtz*
*et al.*, 2014]. While the CPOM and AWI products use a leading edge 50% threshold
retracker, *Kurtz and Harbeck* [2017] rely on a physical model to best fit each CryoSat-2
waveform. This will lead to ice thickness differences based on different thresholds applied:
*Kurtz et al.* [2014] found a 12 cm mean difference between using a 50% threshold and a
waveform fitting method.
We note that several factors contribute to CS2-derived sea ice thickness uncertainties,
including the assumption that the radar return is from the snow/ice interface [*Willat et al.*,
2011], snow depth departures from climatology and the use of fixed snow and ice densities.
In this study we initialize the CICE model simulations described below with the CPOM sea
ice thickness fields.  Accuracy of the CPOM product has been evaluated in several studies,
suggesting mean biases between thickness observations in 2011 and 2012 of 6.6 cm when
compared with airborne EM data [*Laxon et al.*, 2013; *Tilling et al.*, 2015]. For April 2017, the
CPOM near-real-time product [*Tilling et al.*, 2016] was used in place of the archived product,
with a mean thickness bias of 0.9 cm between these products.
In this study, individual thickness point measurements are binned into 5 CICE thickness
categories (1: < 0.6m, 2: 0.6-1.4m, 3: 1.4-2.6m, 4: 2.6-3.6m, 5: > 3.6m) on a rectangular
50km grid for each month. The mean area fraction and mean thickness is derived for each
thickness category and these values are interpolated on the tripolar 1 degree CICE grid
(~40km grid resolution). Grid points with less than 100 individual measurements and a mean
SIT < 0.5 m are not included. For November, this effectively limits the area of the Arctic to
the region shown in Figure 1(c). Negative thickness values that are retained in the CS2
processing to prevent statistical positive bias of the thinner ice are added to category 1. The
novel approach of initializing the CICE model with the full ITD rather than the mean sea ice
thickness provides an additional control on the repartition of the ice among different
thickness categories. This in turn allows a more accurate representation of ice growth and ice
melt processes [*Tsamados et al.*, 2015] compared to initializing with the mean grid-cell SIT
and deriving the fractions for each ice category assuming a parabolic distribution. Ice growth
and melt strongly depend on SIT: using a real distribution can have a big impact, especially
for thin ice.
*CICE Simulations*
CICE is a dynamic-thermodynamic sea-ice model designed for inclusion within a global
climate model. The advantages of using CICE for this study is that we can more readily
separate thickness anomalies into their thermodynamic and dynamical contributions, examine
inter-annual variability and perform longer simulations. For this study, we performed two
different CICE simulations. The first is a multiyear simulation from 1985 to 2017 (referred to
as CICE-free). The second is a stand-alone sea-ice simulation for the pan-Arctic region
starting in mid-November and running until the end of April of the following year for the last
7 winter periods from 2010/2011 to 2016/2017. This results in seven 1-year long simulations
(referred to as CICE-ini), in which the initial thickness and concentration for each of the 5 ice
categories is updated from the CS2 ITD using the CPOM CS2 November thickness fields.



For grid points without CS2 data, and for all other variables (e.g. temperature profiles, snow
volume), results from the free CICE simulation with the same configuration started in 1985
are applied. In this way, CICE simulations cover the pan-Arctic region, but in regions where
no CS2 are available, we restart SIT values from the free CICE model run. While this
approach would be problematic in a coupled model, in a stand-alone sea ice simulation the
model adjustment to the new conditions is smooth and the impact of using the vertical
temperature profile from the free simulation only affects sea ice thickness on the order of
millimeters. While snow accumulation can depart strongly from the *W99* climatology for
individual years, we make the assumption that the deviation of the mean *annual* cycle of
snow depth over the last 7 years from the *W99* climatology is small. Thus, we assume mean
winter ice growth to be determined accurately from CS2, and tuned CICE-ini accordingly to
match the observed CS2 mean winter ice growth from the CPOM product in the central
Arctic [**Figure 1**]. The excellent agreement for both CICE-ini and CICE-free with CS2
increases the confidence of our model results. Our approach therefore allows us to study
inter-annual variability from 2 model configurations with different sources of errors, in
addition to the 3 CS2-based products.
For both CICE simulations, NCEP-2 provides the atmospheric forcing. We use NCEP-2
2m air temperatures because they have been shown to be more realistic for the Arctic Ocean
than those from ERA-Interim [*Jakobshavn et al.*, 2012]. The setup is the same as described *in*
*Schröder et al.* [2014] including a simple ocean-mixed layer model, a prognostic melt pond
model [*Flocco et al.*, 2012] and an elastic anisotropic-plastic rheology [*Tsamados et al.*,
2013], with the following improvements: we apply an updated CICE version 5.1.2 with
variable atmospheric and oceanic form drag parameterization [*Tsamados et al.* 2014], we
increase the thermal conductivity of fresh ice from 2.03 W/m/k to 2.63 W/m/K, snow from
0.3 W/m/K to 0.5 W/m/K and the emissivity of snow and ice from 0.95 to 0.976. While the
default conductivity values are at the lower end of the observed range, the new values are at
the upper end and have been applied in previous climate simulations [e.g. *Rae et al.*, 2014].
Below, all CS2-derived sea ice thickness anomalies are computed relative to the CS2
time-period: November anomalies are relative to 2010-2016, and for April they are relative to
2011-2017. Results for November and April are only shown for all grid cells which have a
minimum thickness of 50 cm and a minimum of 100 individual measurements for each of the
seven years. For the month of November, this corresponds to all colored area shown in Figure
1(c). For April, this region represents the area in red shown in Figure 1(d). The larger region
shown in Figure 1(d) also corresponds to the region over which the amount of
thermodynamic ice growth and dynamical ice growth between November and April are
assessed from the CICE simulations. Further note that area-averaged values for November
and April are only given for regions shown in Figure 1(c) and Figure 1(d), respectively.

## Results

### *Air temperature and freezing anomalies*

The growing season air temperatures anomalies (i.e. mid-November 2016 to mid-April
2017 relative to 1981-2010) were positive throughout the Arctic, leading to large reductions
in the number of FDDs, computed as the cumulative daily 2 m NCEP-2 air temperatures
below -1.8$^{\mathrm{o}}$C, similar to *Ricker et al.* [2016]. FDDs computed this way reflect both the
number of days with air temperatures below freezing, and the magnitude of below freezing
air temperatures over the specified period. Spatially, FDD anomalies show widespread
reductions over most of the Arctic Ocean, with the largest reductions in the Barents and Kara
seas, stretching across the pole towards the Beaufort and Chukchi seas [**Figure 2b**]. In
contrast, during winter 2015/2016, FDDs were most notably anomalous within the Barents
and Kara seas [**Figure 2a**], in agreement with *Ricker et al.* [2017a]. Overall, as averaged



199 from 70-90°N, this past winter witnessed the least amount of cumulative FDDs since at least
200 1979 [**Figure 2c**].
201  While ice forms quickly within the central Arctic once air temperatures drop below
202 freezing, this year saw large delays in freeze-up throughout the Arctic. Updating results
203 previously reported in *Stroeve et al*. [2014], freeze-up was delayed by 20 days for the Arctic
204 as a whole, with regions like the Bering, Beaufort, Chukchi, East Siberian and Kara seas
205 delayed by three to four weeks [**Figure 2d**]. Within the Barents Sea, the regionally averaged
206 freeze-up was delayed by 60 days. In recent years, the trend towards later freeze-up has
207 increased, with the Barents and Chukchi seas showing the largest trends on the order of +14
208 days per decade through 2017, followed by the Kara and East Siberian seas with delays on
209 the order of +10 to +12 days per decade. Within the Beaufort Sea, freeze-up is now
210 happening later by +9 days per decade [**Table 1**].

212 ***November ice thickness anomalies***
213  Before analyzing how the reduced number of freezing degree days impacted winter ice
214 growth during 2016/2017, it is useful to first inter-compare the different CryoSat-2 thickness
215 estimates. We start with a comparison of November thickness from the three CS2 data sets
216 from November 2010 to 2016 [**Figure 3**].
217  It is encouraging to find that year-to-year variability in the spatial patterns of positive and
218 negative thickness anomalies are generally consistent between the three products despite
219 differences in waveform processing. The AWI and CPOM data sets are in better agreement
220 with each other than with the NASA product, which is expected as they use a similar
221 retracker. Furthermore, all three data sets show widespread thinner ice in November 2011,
222 and widespread thicker ice in November 2013. This is further supported by analysis of
223 regional mean thickness and anomalies computed over the region shown in Figure 1(c)
224 [**Table 2**]. For comparison, we also list results from the CICE-free model simulation. In
225 November 2011, the different CS2 data products are in agreement that the ice was
226 anomalously thin (-32 to -46 cm), the thinnest in the CS2 data record. Similarly, in November
227 2013, all three CS2 products show overall thicker ice on the order of +23 to +38 cm. The
228 CICE-free simulations also show anomalously thinner and thicker ice during these years, but
229 larger anomalies were simulated in 2012 and 2014.
230  While the overall pattern of years with anomalously thin or thick ice is broadly similar
231 between the three CS2 products, this is not true in 2016. Both the CPOM and AWI thickness
232 estimates suggest slightly thicker ice than average (+4 cm and +9 cm, respectively), while the
233 NASA product suggests the icepack was overall slightly thinner (-1 cm). The CICE-free run
234 is in agreement with the NASA data set for the 2016 anomaly. Turning back to **Figure 3**, we
235 find that in 2016 the CPOM data set shows +20 to +60 cm thicker ice north of the Canadian
236 Archipelago (CAA) and Greenland, -20 to -60 cm thinner ice on the Pacific side of the pole,
237 and +10 to +30 cm thicker ice north of the Laptev Sea. These spatial patterns of November
238 2016 SIT anomalies are broadly similar with those from AWI but less so with NASA.
239 However, despite similar patterns of positive and negative thickness anomalies, AWI shows
240 between +20 and +30 cm thicker ice over much of the central Arctic Ocean, and even thicker
241 ice (up to +60 cm) north of the CAA and Greenland in November 2016 than the CPOM
242 product. NASA on the other hand shows larger negative anomalies on the Pacific side of the
243 north pole of up to -70 cm and larger positive anomalies directly north of the CAA between
244 +10 and +20 cm.
245  Since we use CPOM CS2 thickness fields to initialize our CICE model runs, this
246 comparison is useful in determining whether or not the 2016 November thickness anomalies
247 are robust in other CS2 processing streams and provides a measure of CS2 sea ice thickness
248 uncertainty.





However, since we do not have the AWI and NASA ITDs we cannot quantify the impact of
using a different thickness data set on our simulations. However, as a result of the negative
winter ice growth feedback (discussed below), differences due to model initialization in
November will be attenuated until April.
***Sea Ice growth from November to April***
For a more robust analysis of winter ice growth during the record warm winter of
2016/2017, we now include April thickness estimates from CS2 (CPOM, AWI and NASA),
the free CICE simulation and the CICE simulations initialized with CPOM CS2 November
SIT in **Figure 4**. Corresponding values for all other years are shown in **Figure 5** (CS2) and
**Figure 6** (CICE). **Table 3** summarizes associated mean April thickness and anomalies since
2011, together with contributions from thermodynamics (ice growth) and dynamics (ice
transport and ridging) based on the CICE model simulations. The area for which these
estimates are provided corresponds to the area shown in Figure 1(d).
We first note that all 5 estimates have different strengths and weaknesses: while the mean
annual cycle of sea ice thickness *should* be more accurate from CS2 than modeled estimates,
robust analysis of winter ice growth from CS2 is in part limited due to the impact of
climatological snow depth assumptions, which may differ from one year to the next, and
differences in waveform processing between CS2 data providers, which may result in
inconsistencies in the magnitude and direction of the observed thickness anomalies. In the
free CICE simulation, November sea ice thickness is less certain due to error accumulation
during the model run. In the initialized CICE simulation, both these error sources are reduced
but inherent model biases remain.
Despite these limitations, all five approaches show good agreement in most years
regarding the direction of the thickness anomalies (i.e. positive or negative) even if they
disagree on absolute magnitude. For example, Arctic Ocean mean thickness anomalies are
negative in all 3 CS2 products for April 2013 (ranging from -3 to -25 cm), whereas in April
2014 and 2015 all approaches give positive mean thickness anomalies, ranging from +5 to
+20 cm in 2014 and +11 to +22 cm in 2015 [**Table 3**]. In some years, the CICE-free
simulation better matches the observed April thickness anomalies (e.g. 2013, 2015), whereas
in other years CICE-ini performs better (e.g. 2012, 2014). On the other hand, in 2011 and
2017 we find disagreement among the three CS2 data sets. In April 2011, both the CPOM
and NASA product have overall negative thickness anomalies for the Arctic Basin (-4 and -8
cm, respectively), whereas they are positive in the AWI product (+7 cm). In April 2017, both
the CPOM and AWI are in close agreement that the ice cover was overall thinner (-13 and -
12 cm, respectively), as are the CICE-free and CICE-ini simulations (negative thickness
anomalies of -13 cm), whereas NASA shows a weak positive anomaly (+3cm).
Focusing more on April 2017, the 3 CS2 products suggest sea ice within the Chukchi and
East Siberian seas was on average -10 to -35 cm thinner in April 2017 compared to the 2011-
2017 mean **[Figure 4(top)]**. CICE simulations show more widespread thinning throughout
the western Arctic, including the Beaufort Sea **[Figure 4(middle and bottom)]**. In the
Beaufort Sea, there is general disagreement among the 3 CS2 products and the CICE
simulations: regional mean anomaly of -5 cm (CPOM), 0 cm (AWI), +20 cm (NASA), -25
cm (CICE-ini) and -30 cm (CICE-free). There is also disagreement north of the CAA, with
CICE-ini indicating positive thickness anomalies (up to +50 cm), whereas all 3 CS2 products
generally show negative thickness anomalies (up to -80 cm). In this region, the CICE-free
simulation also shows mostly negative thickness anomalies (-20 to -80 cm), with a small
positive area (up to +25 cm).
While the discrepancy in this region is puzzling, the bias between the CICE-ini
simulations and the CS2 products in part may reflect the use of a snow climatology in the



CS2 thickness retrievals. As discussed earlier, a positive sea ice thickness anomaly was found
in the November 2016 CS2 thickness retrievals north of CAA and Greenland. Yet this
positive thickness anomaly is not preserved through April in both the CPOM and AWI CS2
products. **Figure 7** shows CICE simulated snow depth anomalies in November 2016 and
April 2017. In November, small positive snow depth anomalies occur throughout the Arctic,
especially north of the Queen Elizabeth Islands where the anomaly locally increases to 20
cm. By April, the anomalies cover a broader region and increase in magnitude. A positive
April snow depth anomaly of 15 to 20 cm relative to *W99* would result in an underestimation
of the CS2-retrieved April ice thickness (SIT) by 79 to 106 cm using the following equation
[*Armitage et al.*, 2015]:

$$SIT = \frac{\rho_{snow} H_{snow} + \rho_{water} F_i}{(\rho_{water} - \rho_{ice})}$$


where we choose snow density ($\rho_{snow}$) of 320 kg/m$^3$ [*Warren et al.*, 1999], ice density ($\rho_{ice}$) of
915 kg/m$^3$, water density of ($\rho_{water}$) 1024 kg/m$^3$. For a radar penetrating to the ice-snow
interface and accounting for the reduced propagation of the speed of light through the snow
cover (2.4 10$^8$ m/s [Tilling et al., 2017]) the ice freeboard ($F_i$) as a function of the radar
freeboard ($F_b$) is $F_i = F_b + 0.25H_{snow}$. CICE-ini, which relies on the CPOM CS2 November
thickness, maintains this positive thickness anomaly through April despite reduced
thermodynamic ice growth. The CICE-free simulation on the other hand started with negative
thickness anomalies in November within this region, and maintains them through April.
One advantage of using CICE, is that we can more readily diagnose thermodynamic vs.
dynamical contributions to the observed thickness anomalies. CICE simulations suggest the
overall thinner ice in April 2017 is largely attributed to reduced thermodynamic ice growth.
One would expect thermodynamic ice growth to be reduced in regions of enhanced snow
depth and thicker November ice. Spatially, the largest reductions in thermodynamic ice
growth during winter 2016/2017 occurred within the Chukchi Sea and north of the CAA and
extending through the northern Beaufort Sea (on the order of -40 cm). These regions have
very different explanations for reduced thermodynamic ice growth. Ice formed a month later
than the 1981-2010 mean within the Chukchi Sea, reducing the number of days over which
the ice could grow. In contrast, north of the CAA, winter ice growth was reduced in a region
that showed positive November thickness anomalies, illustrating the strong dependence of
thermodynamic ice growth on initial ice thickness. This region also had anomalously positive
snow depths that extended through the northern Beaufort Sea, in agreement with extended
regions of reduced thermodynamic ice growth.
While the CICE simulations show reduced thermodynamic ice growth for most of the
Arctic over winter 2016/2017, ice growth was enhanced directly north of Utqiaġvik, Alaska
(formerly Barrow). However, this enhanced ice growth was offset by ice divergence, leading
to overall thinner ice in the CICE simulations. In situ observations of level first-year ice
thickness off the coast of Utqiaġvik ranged between 1.35 and 1.40m during May
(http://arcus.org/sipn/sea-ice-outlook/2017/june) and appear to be in better agreement with
the CICE simulations, as well as the CPOM and AWI CS2 thickness estimates, while the
NASA CS2 product shows positive thickness anomalies in that region. Positive
thermodynamic ice growth anomalies are also found for a small region north of Greenland
and within Fram Strait, as well as within some scattered coastal regions of the Chukchi, East
Siberian, Laptev and Kara seas.
Finally, dynamical thickness changes simulated by CICE show an overall thickening of
the ice in winter 2016/2017 particularly within the Chukchi and Bering seas (up to 50 cm).



Anomalous ridging in this region is in agreement with observed high amounts of deformation
along the shore fast ice zone within the Chukchi Sea as a result of persistent west winds from
December to March (http://arcus.org/sipn/sea-ice-outlook/2017/june). Even larger dynamical
thickening was found within the Kara and northern Barents seas (up to 1.2 m) and to a lesser
extent over the southern and western Greenland Sea, Baffin Bay and the Labrador Sea (not
shown). The CICE-simulated dynamical thickening in the Barents and Kara seas is more
anomalous than seen during previous CS2 years [**Figure 6**], and likely reflects the influence
of the positive Arctic Oscillation (AO) on ice motion [**Figure 8**]. The AO was positive from
December through March, a pattern which results in offshore ice advection from Siberia and
enhanced ice advection through Fram Strait [*Rigor et al.*, 2002]. This pattern leads to
development of thin ice in newly formed open water areas, increasing thermodynamic ice
growth in the Laptev Sea, whereas increased ice advection from thick ice regions north of
Greenland towards Fram Strait, combined with changes in internal ice stress as the ice cover
has thinned, leads to more deformation. Interestingly, while the CICE model runs confirm
overall slightly thinner ice within the Barents Sea in April 2016, consistent with the studies
by *Ricker et al.* [2017a] and *Boisvert et al.* [2016], the thinning from reduced thermodynamic
ice growth was largely offset by thickening from dynamical effects [**Figures 5 and 6**].

*Negative feedbacks*
Ice growth after the September minima is a result of turbulent heat flux exchanges
between the relatively warm ocean mixed layer and the cold autumn and winter air through
the snow-covered sea ice. Progressively, as the ice grows to about 1.5 to 2 m thick, the ocean
becomes well insulated from the atmosphere and ice growth is slowed. Thus, it is not
surprising that we see less thermodynamic ice growth in regions of relatively thick (> 2.5 m)
November ice. A case in point is seen in winter 2013/2014 when thermodynamic ice growth
was reduced by 9 to 10 cm, despite an overall colder winter.
On the other hand, thinner ice regions generally exhibit more vigorous ice growth. For
example, during winter 2012/2013, CICE-free, and to a lesser extent CICE-ini simulated
thermodynamic ice growth increased throughout much of the Arctic Ocean in areas where the
ice retreated in September 2012 [**Figure 6**] and where the November 2012 thickness
anomalies were negative [**Figure 3**]. This process of rapid winter ice growth over thin ice
regions represents a negative feedback, allowing for ice to form quickly over large parts of
the Arctic Ocean following summers with reduced ice cover and thinner November ice.
Thus, while summer sea ice is rapidly declining, several studies have indicated negative
feedbacks over winter continue to dominate [e.g. *Notz and Marotzke*, 2012; *Stroeve and Notz*,
2015], allowing for recovery following summers with anomalously low sea ice extent, such
as those observed in 2007 and 2012. This is further supported in the CICE-free simulations
which show the least amount of winter ice growth in 1989, and peak ice growth following the
2007 and 2012 record minimum sea ice extent [**Figure 9**]. As a result, mean ice growth from
November to April in CICE simulations from 1985 to 2017 shows a positive trend that is
weakly correlated to winter air temperatures or FDDs (*R=0.49*). On the other hand, we find a
strong inverse correlation (*R=-0.82*) between November sea ice thickness and winter ice
growth. Thus, because thin ice grows faster than thick ice, there is an overall stabilizing
effect that suggests as long as air temperatures remain below freezing, even if they are
anomalously warm, the ice can recover during winter. This stabilizing feedback over winter
means that major departures of the September sea ice extent from the long-term trend caused
by summer atmospheric variability generally does not persist for more than a few years
[*Serreze and Stroeve*, 2015].
However, since 2012, overall ice growth has declined as winter air temperatures have
increased further. The correlation from 1985 to 2012 is smaller than over the full record



(*R=0.34*), suggesting a growing influence of warmer winter air temperatures though the
difference in correlation is not statistically significant. While there remains a large amount of
inter-annual variability in winter warming events, *Graham et al.* [2017] suggest a positive
trend in not only the maximum temperature of these warming events, but also in their
duration. Interestingly, there is a modest correlation between detrended FDDs and the winter
maxima sea ice extent (*R=0.30*); not removing the trend results in a correlation of *R=0.83*.
Thus, recent reductions in overall FDDs may have played a role in the last three years of
record low maxima extents.
**Discussion**
407        The CICE-simulations and CS2 thickness retrievals from CPOM and AWI show
consistency that the Arctic sea ice cover in April 2017 was on average 13 cm thinner than the
2011-2017 mean. However, it was likely not the thinnest during the CS2 data record.
Thickness retrievals from the different CS2 data sets showed April 2013 thickness anomalies
were mostly larger than in 2016, ranging from -13 to -25 cm, whereas the CICE simulations
showed much smaller anomalies (-3 to -12 cm). While we expect retrievals from satellite to
be more accurate than those from model simulations, whether or not a year is anomalously
low relative to another year will depend in part on the inter-annual variability in the snow
cover. All three CS2 products rely on the *W99* snow depth climatology. However,
precipitation varies considerably from one year to the next. In the CICE-free simulations,
snow depth is modeled using precipitation from NCEP-2. Inter-annual variability from April
2011 to April 2017 (calculated as standard deviation between the 7 monthly April means) is
shown in **Figure 10**. North of the CAA, standard deviations in snow depth are on the order of
12 to 14 cm, whereas other regions are on the order of 2 to 12 cm. From the *W99*
climatology, inter-annual variability in snow depth during the winter months was estimated to
be only 4 to 6 cm, significantly less than what is exhibited here. Since ice thickness increases
approximately 5 to 6 times the snow depth uncertainty, a 12 to 14 cm uncertainty would lead
to 72 to 83 cm increase in CS2-derived ice thickness. If we average for the area shown in
Figure 1(d), snow depth anomalies ranged from -6 cm to +6 cm, with a corresponding impact
of -41 to +41 cm on thickness.
427        Besides not accounting for inter-annual variability in snow depth, which makes assessing
thickness anomalies from one year to the next less certain, differences in waveform
processing between the three different CS2 products adds further uncertainty. The fact that
the NASA CS2 product is a general outlier compared to the AWI and CPOM products is
further highlighted in **Figure 11**. Across the area considered (e.g. areas in color shown in
Figure 1(c)), the difference between April and the previous November ice thickness is shown
for each CryoSat-2 year. The AWI and CPOM products tend to exhibit positive ice growth
over winter, focused north of Greenland and the CAA and sometimes also across the pole.
The NASA product on the other hand generally shows less ice growth between November
and April in most years, and even no ice growth in some regions. The reasons for this are
unclear, yet interestingly in winter 2016/2017, all three products show more agreement in
regards to thickness decreases that span a broad region north of Greenland and the CAA,
combined with positive increases south of the pole towards the East Siberian and Laptev seas.
440        Finally, how important were the April thickness anomalies in the evolution of the summer
ice cover in summer 2017? Several studies have discussed how thin winter ice may
precondition the Arctic for less sea ice at the end of the melt season as thinner ice melts and
open water areas form more readily in summer, enhancing the ice albedo feedback [e.g.
*Stroeve et al.*, 2012; *Perovich et al.*, 2008], and sea ice thickness has been used as a predictor
for the September sea ice extent [*Kimura et al.,* 2013]. Thus, we may have expected 2017 to
be among the lowest recorded sea ice extents as the ice cover was likely thinner than average



and the winter extent was the lowest in the satellite record. Nevertheless, the minimum extent
ended up as the 8[th] lowest in the satellite data record. This highlights the continuing
importance of summer weather patterns in driving the September minimum. Spring and
summer 2017 were dominated by several cold core cyclones, leading to near average air
temperatures and ice divergence [see http://nsidc.org/arcticseaicenews/ for a discussion of
this summer's weather patterns]. Overall, the correlation between detrended winter sea ice
thickness anomalies and September sea ice extent remains low [*Stroeve and Notz*, 2015].
Other factors such as melt pond formation in spring [*Schröder et al.*, 2014] and summer
weather patterns still largely govern the evolution of the summer ice pack at current thickness
levels [e.g. *Holland and Stroeve*, 2011]. Interestingly, predictions of the monthly mean
September 2017 sea ice extent based on spring melt pond fraction in May gave a value of 5.0
$\pm$ 0.5 million km$^2$, whereas the observed value was 4.80 million km$^2$ [See arcus.org/sipn/sea-
ice-outlook/2017/june].

## Conclusions

In this study we examined sea ice thickness anomalies derived from three different CS2
data products and that simulated using CICE. Overall freezing degree days were much
reduced in winter 2016/2017, and subsequent sea ice thickness estimates from CryoSat-2 in
April 2017 suggest the ice was thinner over large parts of the Arctic Ocean. These results are
complimented with CICE model simulations, both with and without initializing with
November ice thickness distributions. While CICE simulations suggest the mean thickness
within the Arctic Basin in April 2017 was the thinnest over the CryoSat-2 data record,
corresponding CS2-derived sea ice thickness from the three different data providers put this
into question. However, the use of CS2-derived freeboards with a snow depth climatology
remains problematic because it fails to capture inter-annual snow accumulation variability
which remains a large source of error in current CS2 thickness retrievals. Differences in
processing of the radar waveform, values of snow and ice density, delineation of first-year vs.
multiyear ice, and sea surface height retrieval also contribute to differences among available
data sets, making it challenging to robustly assess inter-annual variability of ice thickness
from CryoSat-2. Despite these challenges it is encouraging that in most years, the interannual
variability in positive and negative anomalies is consistent between the 3 CS2 data sets.
Finally, CICE-free simulations from 1985 to 2017 reveal the correlation between winter
ice growth and November ice thickness ($R=-0.82$) is stronger than between growth and FDDs
($R=0.49$), highlighting the importance of the negative winter growth feedback mechanism.
This supports previous studies that the long-term sea ice reduction in the Arctic Basin is
mainly driven by summer atmospheric conditions. However, this correlation has become
weaker since 2012, indicating that higher winter air temperatures and further delays in
autumn/winter freeze-up due to warmer mixed-layer ocean temperatures prohibit a complete
recovery of winter ice thickness in spite of the negative feedback mechanism. This is
highlighted by the fact that overall thermodynamic ice growth for winter 2016/2017 was just
under 1m despite 2016 reaching the second lowest minimum extent recorded during the
satellite record.

## Acknowledgements

This work was in part funded under NASA grant NNX16AJ92G (Stroeve). Sea ice
simulations and CryoSat-2 satellite data processing performed under NERC funding of the
Centre for Polar Observation and Modeling (CPOM). CryoSat-2 thickness fields courtesy of
A. Ridout at CPOM. Processing of the AWI CryoSat-2 (PARAMETER) is funded by the
German Ministry of Economics Affairs and Energy (grant: 50EE1008) and data from
November 2010 to April 2017 obtained from http://www.meereisportal.de (grant: REKLIM-





2013-04). NASA CryoSat-2 data provided courtesy of Nathan Kurtz. NCEP2 data obtained
from NOAA Earth System Research Laboratory
(http://www.esrl.noaa.gov/psd/data/gridded/data.ncep.reanalysis2.gaussian.html).
Data policy: data available upon request.

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

penetration into snow cover Arctic sea ice using airborne data, *Ann. Glaciol.*, **52**(57), 197–
633      205.





**Table 1**. Regional trends in freeze-up, 2017 freeze-up date and anomaly (relative to 1981-
2010 mean).

| Region | Freeze-up Trend (days per decade) | 2017 Mean Freeze-up (day of year) | 2017 Freeze-up Anomaly (days) |
|---|---|---|---|
| Sea of Okhotsk | 9.1 | 304 | 0.8 |
| Bering Sea | 6.7 | 338 | 25.2 |
| Hudson Bay | 7.9 | 333 | 16.9 |
| Baffin Bay | 8.0 | 312 | 13.2 |
| E. Greenland Sea | 5.6 | 267 | 2.7 |
| Barents Sea | 13.6 | 347 | 60.3 |
| Kara Sea | 10.7 | 314 | 36.6 |
| Laptev Sea | 9.0 | 272 | 10.7 |
| E. Siberian Sea | 11.8 | 286 | 27.1 |
| Chukchi Sea | 14.1 | 314 | 31.0 |
| Beaufort Sea | 8.9 | 279 | 23.4 |
| Canadian Archipelago | 4.9 | 268 | 12.7 |
| Central Arctic | 3.1 | 255 | 16.8 |
| Pan-Arctic | 7.5 | 288 | 19.6 |

**Table 2.** Mean November ice thickness and anomaly with respect to the 2011-2017 mean (in
parenthesis) from CS2 derived from CPOM, AWI and NASA. Spatial mean is over Arctic Basin,
defined as the area for which CS-data were available continuously for all 7 winter periods
November to April 2010/2011 to 2016/17. This region corresponds to all three regions
shown in Figure 1(c).

| | November SIT CS2 CPOM (cm) | November SIT CS2 AWI (cm) | November SIT CS2 NASA (cm) | November SIT CICE-free (cm) |
|---|---|---|---|---|
| 2010 | 183 (-6) | 208 (-8) | 198 (-7) | 206 (+6) |
| 2011 | 157 (-32) | 174 (-42) | 170 (-35) | 185 (-15) |
| 2012 | 173 (-16) | 192 (-24) | 177 (-28) | 152 (-48) |
| 2013 | 212 (+23) | 246 (+29) | 243 (+38) | 208 (+08) |
| 2014 | 207 (+18) | 239 (+23) | 226 (+21) | 231 (+31) |
| 2015 | 196 (+7) | 229 (+13) | 217 (+12) | 219 (+19) |
| 2016 | 193 (+4) | 225 (+9) | 204 (-1) | 199 (-1) |
| 2010-2016 mean | 189 | 216 | 205 | 200 |






**Table 3.** Mean April sea ice thickness (SIT) and anomaly with respect to the 2011-2017 mean (in parenthesis) from three CS2 products (CPOM, AWI and NASA), and the CICE (free run 1985-2017) and CICE runs initialized with CS2 ice thickness in November. The amount of thermodynamic ice growth and dynamical ice change from the CICE model runs is also given. Spatial mean is over Arctic Basin, defined as the area shown in Figure 1(d).

| | CryoSat-2 Results | | | CICE Simulations | | | | | |
|---|---|---|---|---|---|---|---|---|---|
| | April SIT CPOM (cm) | April SIT AWI (cm) | April SIT (NASA) (cm) | April SIT CICE free (cm) | April SIT CICE ini (cm) | Therm growth CICE free (cm) | Therm growth CICE ini (cm) | Dyn change CICE free (cm) | Dyn change CICE ini (cm) |
| 1990-2017 Mean | n/a | n/a | n/a | 283 | n/a | 107 | n/a | -18 | n/a |
| 2010-2017 Mean | 243 | 230 | 235 | 246 | 240 | 112 | 103 | -15 | -17 |
| 2011 | 239 (-4) | 237 (+7) | 227 (-8) | 242 (-4) | 241 (+1) | 115 (+3) | 104 (+1) | -18 (-3) | -20 (-3) |
| 2012 | 235 (-8) | 219 (-11) | 218 (-17) | 247 (+1) | 233 (-7) | 115 (+3) | 110 (+7) | -9 (+6) | -12 (+5) |
| 2013 | 230 (-13) | 208 (-22) | 210 (-25) | 234 (-12) | 237 (-3) | 136 (+24) | 117 (+14) | -16 (+1) | -19 (-2) |
| 2014 | 261 (+18) | 250 (+20) | 254 (+19) | 251 (+5) | 249 (+9) | 102 (-10) | 94 (-9) | -12 (+3) | -17 (+0) |
| 2015 | 264 (+21) | 252 (+22) | 254 (+19) | 264 (+18) | 255 (+11) | 108 (-4) | 103 (-0) | -18 (-3) | -22 (-5) |
| 2016 | 239 (-4) | 227 (-3) | 228 (-7) | 254 (+8) | 241 (+1) | 107 (-5) | 101 (-2) | -15 (-0) | -17 (+0) |
| 2017 | 230 (-13) | 218 (-12) | 238 (+3) | 233 (-13) | 227 (-13) | 99 (-13) | 92 (-11) | -14 (+1) | -13 (+4) |





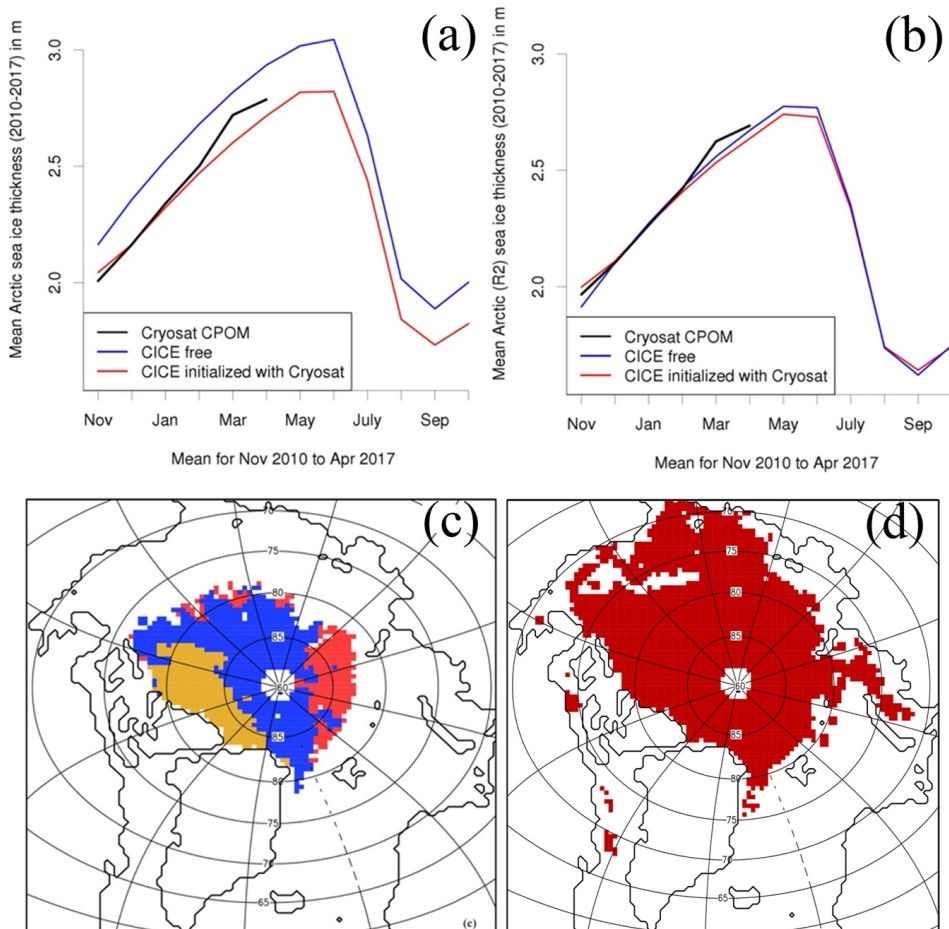

**Figure 1.** Comparison of CPOM CryoSat-2 mean seasonal sea ice thickness (black) with CICE
free (blue) and CICE initialized with Cryosat-2 in November (red). Figure 1(a) shows results
for mean thickness averaged over all the colored areas shown Figure1(c), representing the
total region for which Cryosat-2 data exist in November (only grid points included with > 100
measurements per month and mean thickness > 0.5m) and (b) mean thickness averaged
over the sub-region shown in blue with medium thick ice in January (between 1.5 and
2.5m). Blue areas in Figure 1(c) show regions between November and January where
CryoSat-2 thickness are between 1.5 and 2.5 m in all years. Figure 1(d) is the region over
which the April thickness anomalies and results are presented.





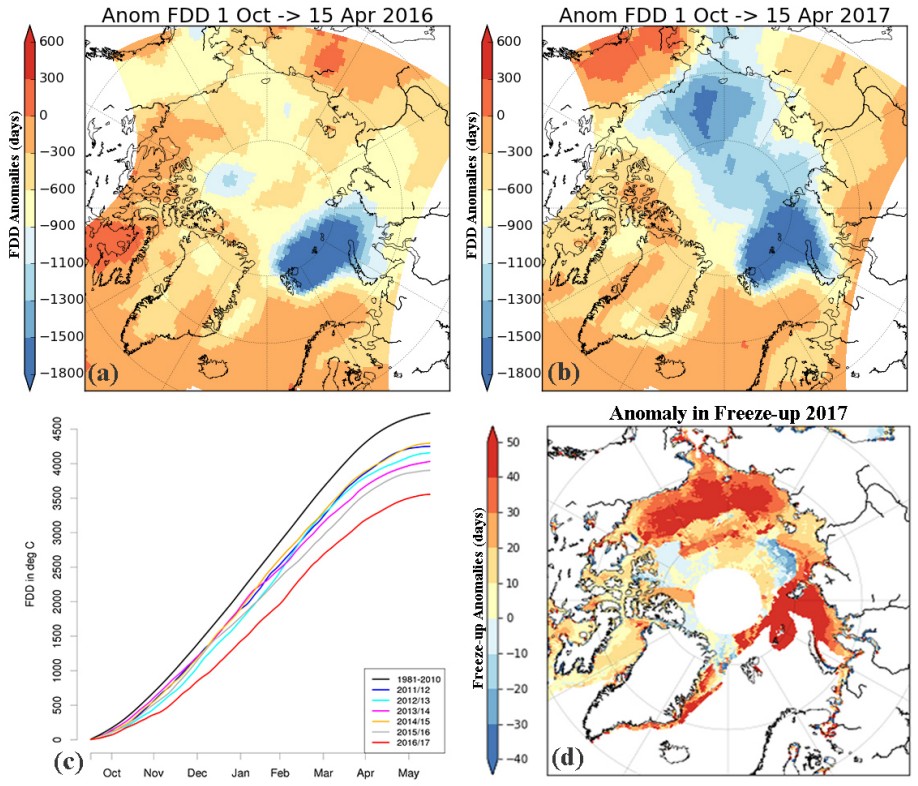

**Figure 2.** Top panel shows the freezing degree anomalies (FDD) computed as the number of days with NCEP2 2m air temperature below -1.8°C from mid-November to mid-April in winter 2016 (a) and winter 2017 (b) computed relative to the 1981-2010 climatology. Bottom left image shows the cumulative freezing degree days (FDDs) averaged over region shown in Figure 3 inset (c), and bottom right image shows freeze-up anomalies for 2016/2017 relative to 1981-2010 (d). Areas in white are either missing (pole hole) or no sea ice in winter 2016/2017. Light gray areas are open ocean.



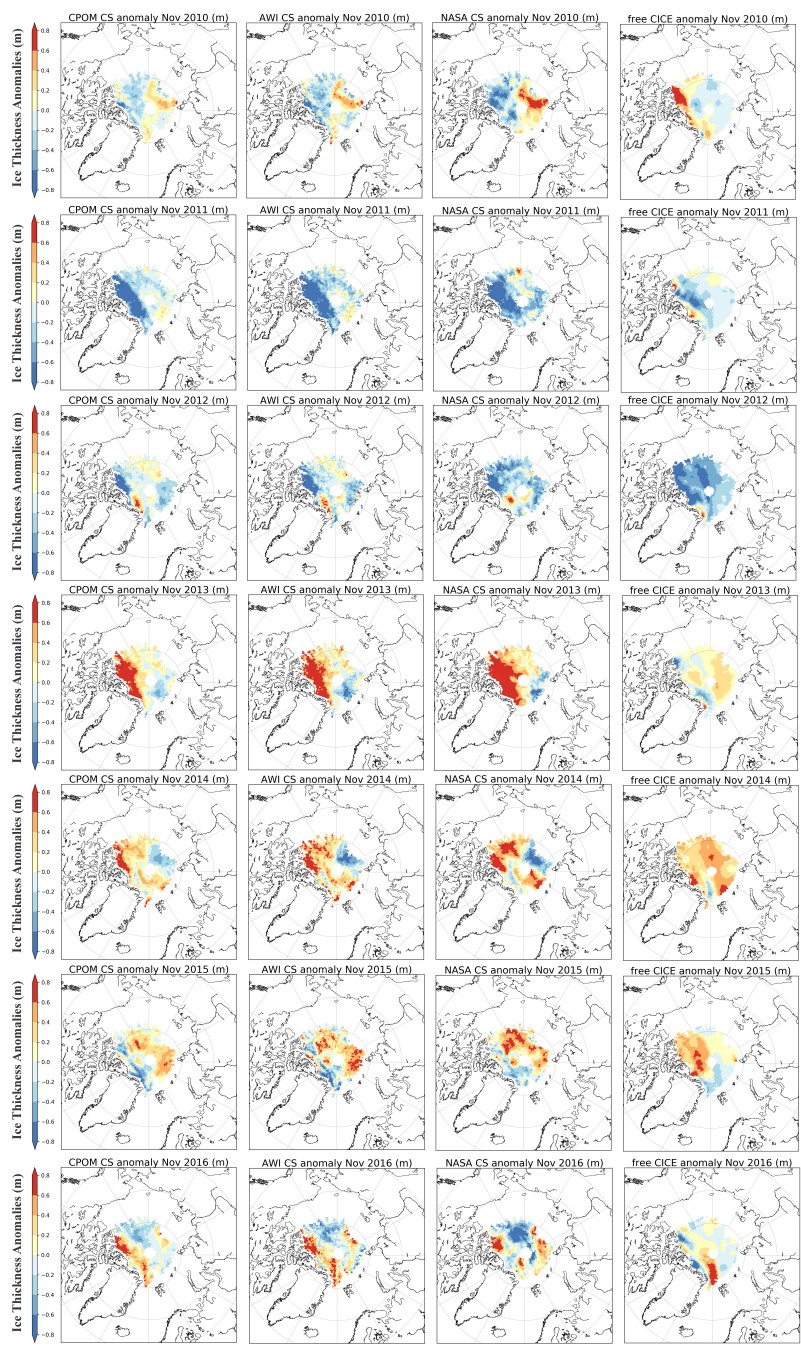

**Figure 3**. November ice thickness anomaly relative to 2010-2016 in cm based on CryoSat-2
data from UCL CPOM (left), Alfred Wegener Institute (AWI) (middle) and NASA (right). Grid
points with less than 100 individual measurements and a mean sea ice thickness of less than
0.5 m are not included. CICE-free thickness anomalies are also shown in the left right
column.




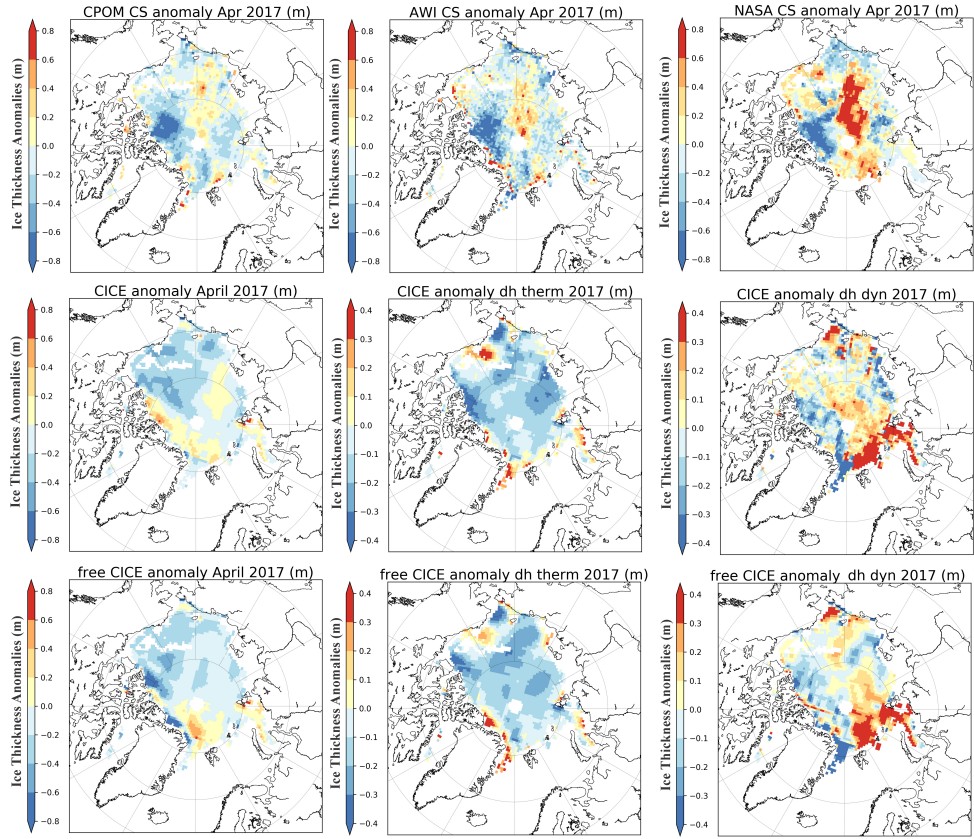

**Figure 4.** CryoSat-2 and CICE simulated thickness anomalies in April 2017 relative to the
2011-2017 mean. Top images show the total ice thickness anomalies from CryoSat-2 for
CPOM (left), AWI (middle) and NASA (right). The middle left image shows April 2017
thickness anomalies from CICE initialized with CPOM November CS2 thickness together with
the contributions from thermodynamics (middle) and dynamics (left) and bottom show the
corresponding results from the CICE free simulations. Grid points with less than 100
individual measurements and a mean sea ice thickness of less than 0.5 m are not included.





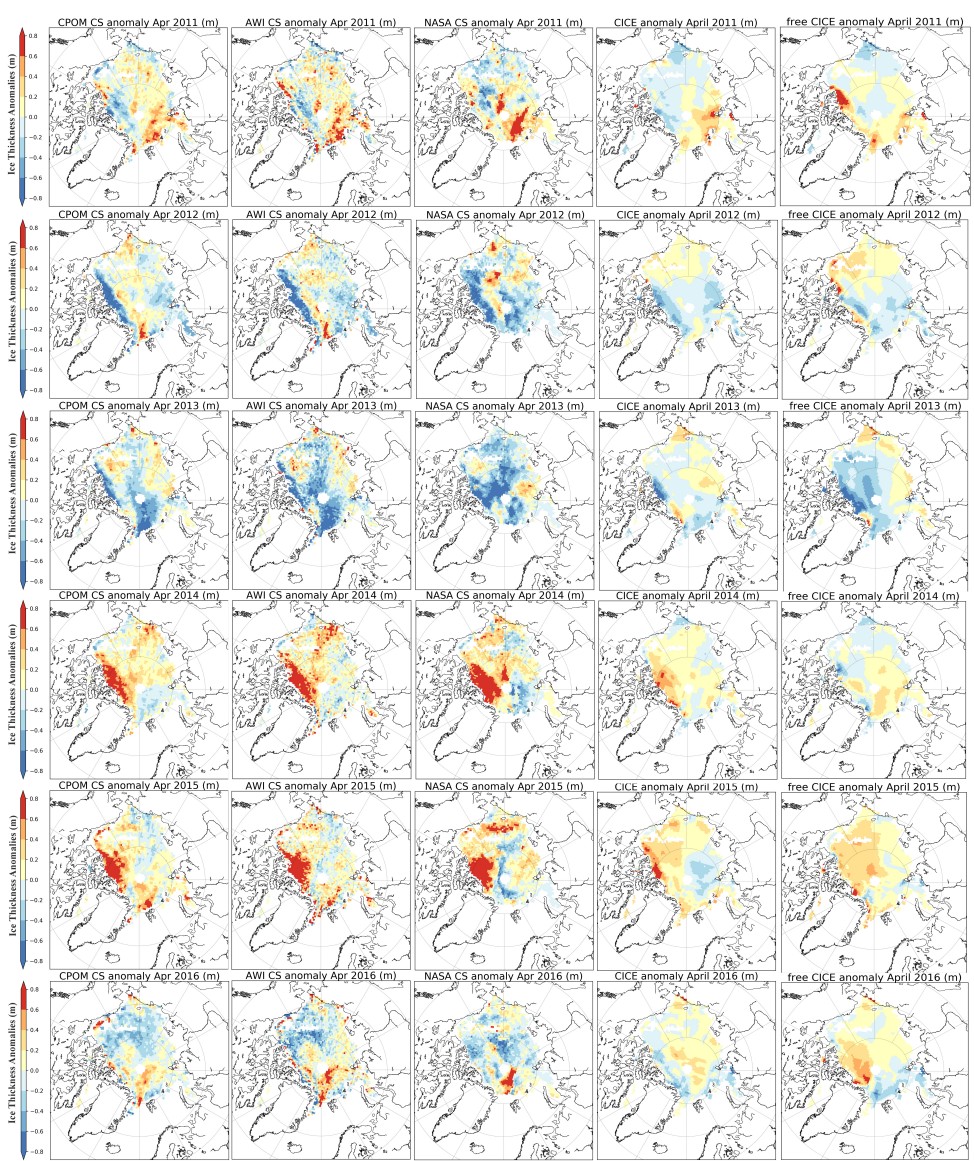

**Figure 5.** Anomaly of April ice thickness from 2011 to 2016 in m relative to the 2011 to 2017 mean from CryoSat-2 CPOM (far left), AWI (second left), NASA (middle), CICE simulations initialized with November CPOM CryoSat-2 thickness fields (2$^{nd}$ right), and CICE simulations not initialized with CryoSat-2 thickness (right). Grid points with less than 100 individual measurements and a mean sea ice thickness of less than 0.5 m are not included.





**Figure 6.** Anomalies of CICE simulated thermodynamic ice growth and dynamical thickness changes in m relative to the 2011 to 2017 mean from the CICE simulations initialized with November CPOM CryoSat-2 thickness fields (left), and CICE simulations not initialized with CryoSat-2 thickness (right). The year in title reflects the end month over which ice growth occurs (e.g. from November to April).





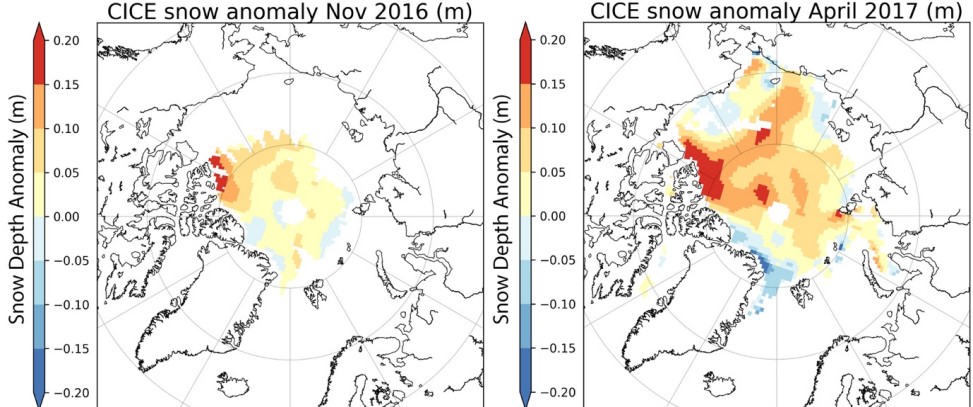

**Figure 7**. Snow depth anomaly for November 2016 (relative to 2010-2016) and April 2017
(relative to 2011-2017) from CICE.





**Figure 8.** Mean monthly sea ice motion from the NSIDC Polar Pathfinder Data Set.
Preliminary data provided by Scott Stewart, NSIDC.





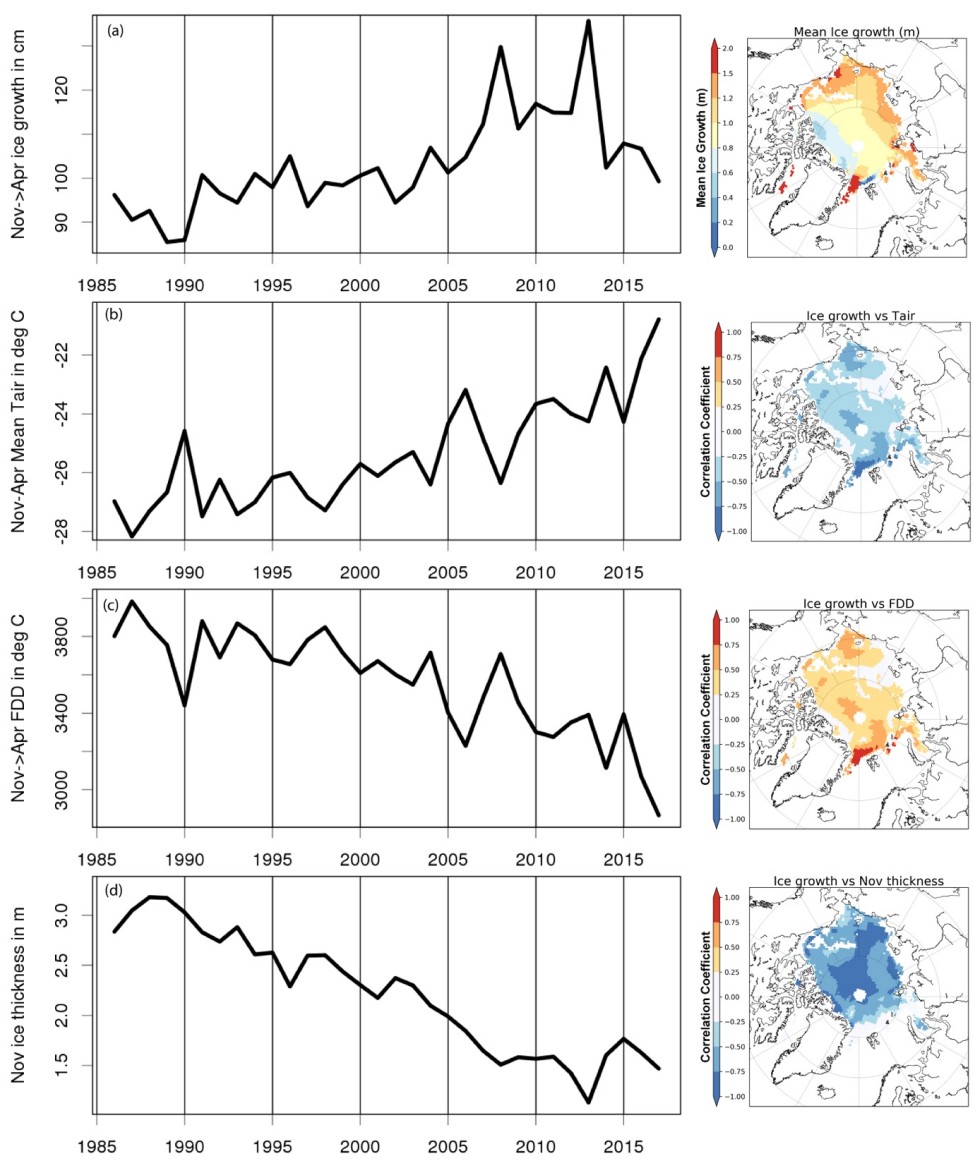

**Figure 9.** Time-series from 1985 to 2017 of mean winter ice growth (mid-November to mid-April) in the free CICE simulation (a), mean 2m NCEP-2 air temperature (b), cumulative freezing degree days (FDDs) (c) and November ice thickness (d). All time-series results are averaged over the areas shown in Figure S1(c). Corresponding images to the left of each time-series plots show: mean ice growth from November to April as averaged from 1985/1986 to 2016/2017; correlation coefficient between ice growth and 2m NCEP-2 air temperature; correlation coefficient between ice growth and FDDs; and correlation coefficient between ice growth and November ice thickness, respectively. All correlation values are given for linear regression of de-trended time series.





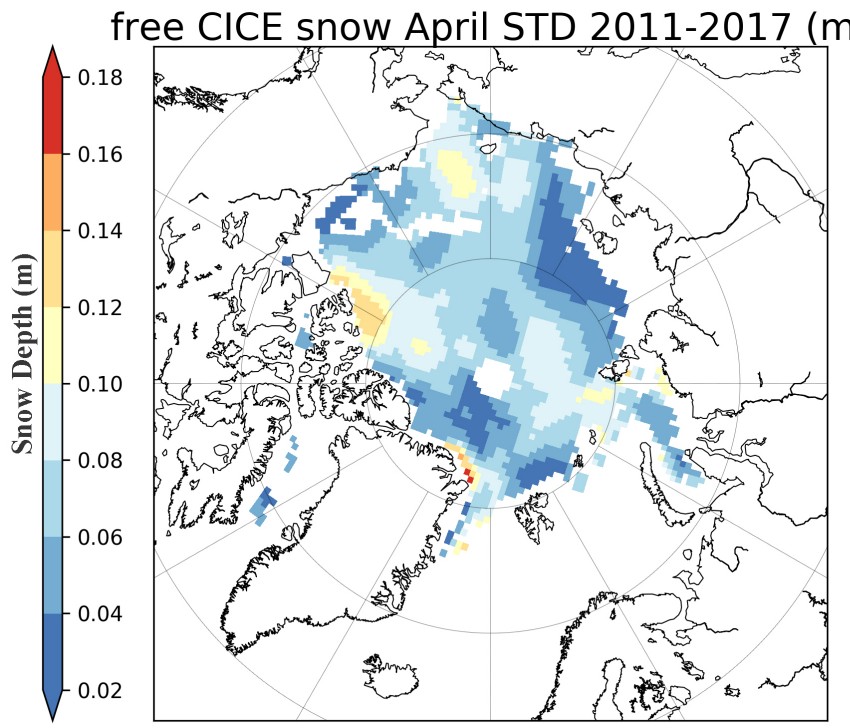

**Figure 10.** Standard deviation of CICE-simulated snow depth using NCEP-2 reanalysis for the month of April from 2011 to 2017.



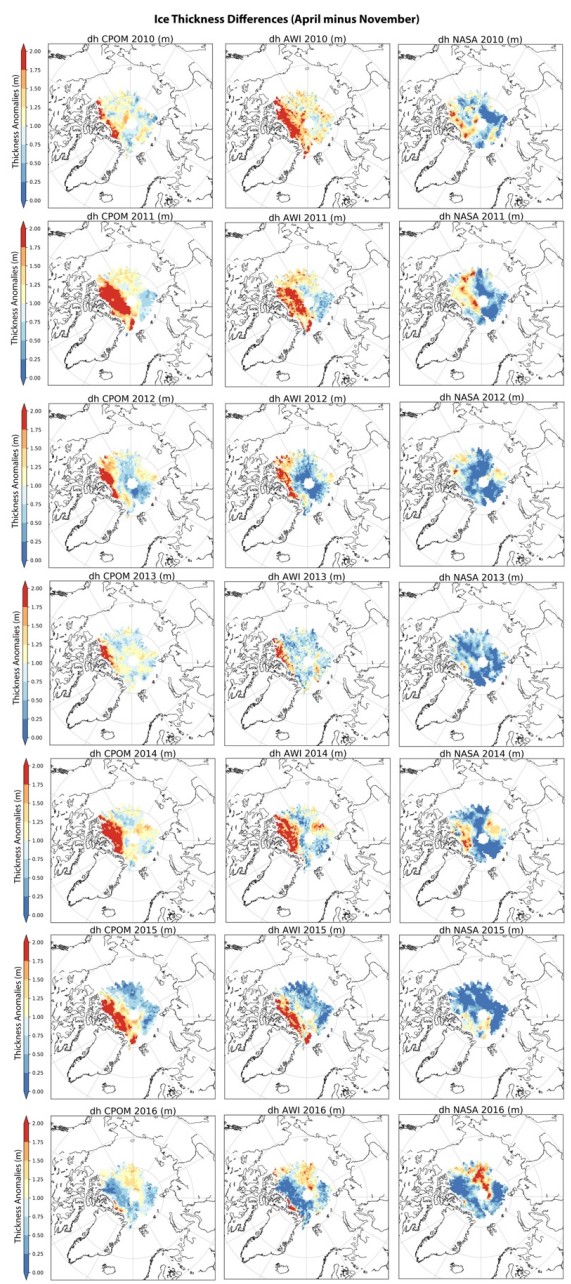

**Figure 11**. Comparison between ice growth (April minus November) in the UCL CPOM
CryoSat-2 thickness retrievals (left) and those from the Alfred Wegener Institute (AWI)
(middle) and NASA (right). The year shown corresponds to the November months, such that
2016 refers to ice thickness differences between April 2017 and November 2016. Results are
only shown for the area shown in Figure 1(c), which represents grid points that had more
than 100 individual measurements and a mean sea ice thickness greater than 0.5 m during
the November months.