# Peer review of "Warm Winter, Thin Ice?"

_The Cryosphere, 2017_

## Referee Comment (RC1) · Anonymous Referee #1 · 3 Feb 2018

Warm Winter, Thin Ice by Stroeve and others

Summary: Stroeve and others investigate the impact of 2016/2017 anomalously warm winter on sea ice thickness using the CICE model and CS2 thickness observations. A secondary objective of the study is to compare three difference approaches of ice thickness retrievals from CS2 to CICE. The authors demonstrate that recent warm fall temperatures (i.e. since 2012) impact winter sea ice thickness by reducing wintertime growth which was particularly strong in 2016/2017. Overall, I think this manuscript can find a place in the literature when the author's address my major concern that thinning in 2016/2017 especially, north of Greenland and the Canadian Archipelago was not entirely driven by thermodynamics (i.e. positive snow depth anomalies) but rather reduced ice convergence.

Major comment: The authors have not made a convincing argument that snow depth is

the primary mechanism for reduced ice thickness north of Greenland and the Canadian Archipelago in April 2017. While I agree snow depth is the major source of uncertainty in CS2 retrievals, ice dynamics during the winter of 2017 in this region was likely more influential and should be discussed. The authors suggest the positive ice thickness anomaly in November 2016 north of Greenland and the Canadian Archipelago did not persist because of snow loading and in turn reduced thermodynamic growth but ice dynamics (i.e. lack of ice convergence) is more likely the culprit here. Indeed, the fall of 2016 was the warmest on record and these temperature anomalies persisted into 2017, thinning ice in some regions (Barents Sea) but this thinning also manifested enhanced surface heating changing atmospheric circulation over the Arctic and especially over the Beaufort Sea. Consequently, the Beaufort High collapsed in the winter of 2017 and this reduced ice convergence against the northern Canadian Archipelago and Greenland which is clearly apparent from the sea ice motion vectors in Figure 8 of the author's paper. The latter process seems to be more likely the cause of why the November ice thickness anomaly in this region was not preserved as atmospheric circulation prevented dynamic ice growth (convergence) which typically dominates during the winter in this region. I think the authors should acknowledge that ice thinning in the Arctic is not entirely thermodynamically driven and ice dynamics also play a role which is underscored by Kwok, 2015, GRL.

A second related point is that multi-year ice is the dominant ice type north of Greenland and the Canadian Archipelago which has consistently been preserved despite the shift from multi-year ice to first-year ice elsewhere in the Arctic. This suggests that the snow depth here should be somewhat similar to the Warren Climatology. This was actually reported to be the case based on recent measurements from Haas et al., 2017, GRL and hence CS2 estimates in this thick MYI region should be reliable. The latter point also lends further support to reduced ice convergence being more influential on thinning than thermodynamics.

Specific Comments Line 286-288 Ok, but there appears to be a mix of positive and neg-

ative anomalies. The most prominent feature worth mentioning is the CS2 strongest thinning anomalies are along the northern coast of the Canadian Archipelago.

Line 297-299 I'm not convinced that the snow loading in CS2 has caused this difference in April 2017 north of the Canadian Archipelago and Greenland. If I recall, the Beaufort High collapsed in the winter of 2017 and this reduced convergence against the northern Canadian Archipelago and Greenland which appears to the case in Figure 8. The latter seems more likely the cause of why the thickness anomaly in this region was not preserved as atmospheric circulation prevent dynamic ice growth. This seems to be captured across all CS2 products but not CICE-ini. This needs revision. See major comment.

Line 413-415 The snow is important but ice thickness is strongly influence by dynamics (i.e. convergence against the Canadian Archipelago and Greenland) and this needs to be mentioned in the discussion as well. See Kwok, 2015, GRL. Furthermore, MYI is the dominant ice type north of Greenland and the Canadian Archipelago which has consistently been preserved despite the shift from MYI to FYI elsewhere. This suggests the snow depth here should be similar to the W99 which was found reported by Haas et al., 2017, GRL hence CS2 estimates here should be reliable and lends further support to reduced ice convergence was more influential on thinning. See major comment.

Table 1 What is the source of the data in this table? The passive microwave algorithm from Markus et al., 2009, JGR?

References: Haas, C., Beckers, J., King, J., Silis, A., Stroeve, J., Wilkinson, J., Notenboom, B., Schweiger, A., & Hendricks, S. (2017). Ice and snow thickness variability and change in the high Arctic Ocean observed by in situ measurements. Geophysical Research Letters, 44, 10,462–10,469. https://doi.org/10.1002/2017GL075434

Kwok, R. (2015), Sea ice convergence along the Arctic coasts of Greenland and the Canadian Arctic Archipelago: Variability and extremes (1992–2014), Geophys. Res. Lett., 42, 7598–7605, doi:10.1002/2015GL065462.

---

## Referee Comment (RC2) · Anonymous Referee #2 · 7 Feb 2018

**Summary**

This paper uses model simulations from the Los Alamos sea-ice model (CICE) and CryoSat-2 thickness estimates from three different data providers to investigate the impact of the 2016/2017 anomalously warm winter on Arctic sea ice thickness. The authors consider free CICE simulations as well as CICE simulations initialized with CryoSat. Coinciding with the least amount of freezing degree days north of 70°N since 1979, the authors find that CICE simulations in April 2017 show the thinnest ice cover in the Arctic Basin over the CryoSat-2 data period. However, this finding is not entirely supported by the satellite retrievals. CICE simulations are also used to investigate the processes leading to ice thickness anomalies, separating dynamic and thermodynamic contributions. It is concluded that free CICE simulations from 1985 to 2017 reveal that

the correlation between winter ice growth and November ice thickness is stronger than between growth and FDDs, although this correlations has become weaker since 2012, and delayed freeze up due to warmer winter temperatures play a bigger role.

General comments:

The impact of warmer winter seasons on the Arctic ice cover is of high interest for the sea ice and climate science community. In addition, the comparison between sea ice thickness retrievals from different providers adds some valuable information here. The manuscript itself is well written, but there are lots of information in the figures and tables which are not easy to capture. For example, color bars in Figure 4 show different scales, which is a bit confusing. Also the quality of the figures in general can be improved. See more detailed comments below.

Apart from that, my major concern is that it is not really well explained how reliable the model simulations are, both CICE free and CICE initialized with CryoSat. Although the mean monthly values seem to fit quite well to the satellite observations, considering Figure 3 and Figure 5, regional anomalies disagree quite significant in several cases. For example, the significant positive thickness anomaly north of the Canadian Archipelago in April 2014 and 2015 is rather weak in the model simulations. I don't think that this is due to the usage of a snow climatology in the satellite retrievals, since this area is mostly covered by multiyear sea ice. I also wonder why this strong positive anomaly is not present at least in the CICE simulations initialized with CryoSat. Based on these concerns, I also wonder how reliable the findings and conclusions regarding the results presented in Figure 9 are. Could you include the satellite observations here as well? Also difference maps and scatter plots between simulated ice thicknesses and CryoSat ice thicknesses would be interesting and could potentially help to support the conclusions and show more explicit the limitations of the model simulations. For example, how meaningful are the correlations given in the maps of Figure 9 if the model is limited in reproducing regional anomalies as described above?

Detailed comments:

P3 L109: The CPOM product is derived using a 70 % threshold, not 50 % as stated in this paper (and in Laxon et al. (2013) because of a typo). There is an erratum for Laxon et al. (2013) where a 70% threshold is reported.

P3 L124: Category 1 ranges up to 0.6 m. But when you discard any measurements below 0.5 m, then you this category only covers a very narrow range of thickness. Isn't that a limitation for the initialization of the model then?

P3 L138: CICE simulations - What are the grid cell ice thicknesses in the CICE simulations representing? The mean thickness of the ice covered area or the mean thickness of the entire area including open water? This information should be given in this section, because it is crucial when comparing it with the satellite data.

Figure 1 c): Information about the red and the yellow areas is missing.

Figure 2, L677: I cannot see any light gray areas. The legend in Fig 2c is very small.

Figures 3, 5, 6, 11: The labels of the color tables are too small. Since all maps of each figure correspond on the same thickness range, I suggest to use just one color bar and make it bigger.

Figure 4: It is a bit confusing that you use different thickness ranges for the CICE anomaly contributions from thermodynamics and dynamics (+/- 0.4), while for the other maps, you use +/- 0.8. I suggest to use a uniform range, e.g. +/- 0.8. This would make a comparison with the other maps easier.

Second, I wonder how to interpret the thermodynamic and dynamic contributions. For example, there is a positive CICE anomaly north of the archipelago (middle left), while both the thermodynamic (middle center) and dynamic (middle right) contributions show negative anomalies. How is this explained?

Moreover, there is a typo in the caption (L692). I suppose contribution of dynamics is

shown in the "right" column.

---

## Author Comment (AC1) · 27 Feb 2018

Summary: Stroeve and others investigate the impact of 2016/2017 anomalously warm winter on sea ice thickness using the CICE model and CS2 thickness observations. A secondary objective of the study is to compare three difference approaches of ice thickness retrievals from CS2 to CICE. The authors demonstrate that recent warm fall temperatures (i.e. since 2012) impact winter sea ice thickness by reducing wintertime growth which was particularly strong in 2016/2017. Overall, I think this manuscript can find a place in the literature when the author's address my major concern that thinning in 2016/2017 especially, north of Greenland and the Canadian Archipelago was not entirely driven by thermodynamics (i.e. positive snow depth anomalies) but rather reduced ice convergence.

**We thank the reviewer for their comment. We agree with the reviewer that the anomaly in 2016/2017 was not entirely driven by thermodynamics and thus it is a fair point that we should have discussed in more detail. In response we have now stated more explicitly the role that dynamics also played in reducing the ice thickness north of CAA. We actually already showed this in our model results (strong negative dynamical thickness reduction for 2017 in Figure 4 from CICE and also the free-CICE simulation as well as by the ice motions in Figure 8. Thus, we have now made this point clearer in our discussion of the results. We appreciate the reviewer pointing out our need to expand on this discussion.**

Major comment: The authors have not made a convincing argument that snow depth is the primary mechanism for reduced ice thickness north of Greenland and the Canadian Archipelago in April 2017. While I agree snow depth is the major source of uncertainty in CS2 retrievals, ice dynamics during the winter of 2017 in this region was likely more influential and should be discussed.

**See our comments above.**

The authors suggest the positive ice thickness anomaly in November 2016 north of Greenland and the Canadian Archipelago did not persist because of snow loading and in turn reduced thermodynamic growth but ice dynamics (i.e. lack of ice convergence) is more likely the culprit here. Indeed, the fall of 2016 was the warmest on record and these temperature anomalies persisted into 2017, thinning ice in some regions (Barents Sea) but this thinning also manifested enhanced surface heating changing atmospheric circulation over the Arctic and especially over the Beaufort Sea. Consequently, the Beaufort High collapsed in the winter of 2017 and this reduced ice convergence against the northern Canadian Archipelago and Greenland which is clearly apparent from the sea ice motion vectors in Figure 8 of the author's paper. The latter process seems to be more likely the cause of why the November ice thickness anomaly in this region was not preserved as atmospheric circulation prevented dynamic ice growth (convergence) which typically dominates during the winter in this region. I think the authors

should acknowledge that ice thinning in the Arctic is not entirely thermodynamically driven and ice dynamics also play a role which is underscored by Kwok, 2015, GRL.

**We agree with the reviewer (as noted in our comments above) and we have discussed this more extensively in the revised version.**

A second related point is that multi-year ice is the dominant ice type north of Greenland and the Canadian Archipelago which has consistently been preserved despite the shift from multi-year ice to first-year ice elsewhere in the Arctic. This suggests that the snow depth here should be somewhat similar to the Warren Climatology. This was actually reported to be the case based on recent measurements from Haas et al., 2017, GRL and hence CS2 estimates in this thick MYI region should be reliable.

**While I was a co-author on the Haas et al. paper, I disagree with the assertion that we should expect each year the snow depth to be on the same order as climatology. Snow depth varies considerably from year to year. In fact, we find in the reanalysis data used in the CICE simulations that there are years with anomalously high and low snow accumulation which is illustrated in Figure 10. Regions with the largest standard deviation are actually north of the CAA. In addition, the figure below shows the interannual variability of Arctic precipitation from 5 different reanalysis, which clearly shows large interannual variability. Thus, we cannot conclude that snow depth anomalies do not play a role in year-to-year sea ice thickness variability in the currently processed CS2 data products.**

[Figure]

The latter point also lends further support to reduced ice convergence being more influential on thinning than thermodynamics.

Specific Comments

1. Line 286-288 Ok, but there appears to be a mix of positive and negative anomalies. The most prominent feature worth mentioning is the CS2 strongest thinning anomalies are along the northern coast of the Canadian Archipelago.

   **Made changes as suggested by the reviewer. Below is the new paragraph:**

   *Focusing more on April 2017, the 3 CS2 products suggest widespread thinner ice in April 2017 north of Ellesmere Island (up to -80 cm thinner) relative to the 2011-2017 mean [**Figure 4(top)**]. Thinner ice is also found within the Chukchi and East Siberian seas (on average -10 to -35 cm thinner) despite a mix of positive and negative anomalies. CICE simulations on the other hand show more widespread thinning throughout the western Arctic, including the Beaufort Sea and positive thickness anomalies north of Ellesmere Island [**Figure 4(middle and bottom)**]. In the Beaufort Sea, there is general disagreement among the 3 CS2 products and the CICE simulations: regional mean anomaly of -5 cm (CPOM), 0 cm (AWI), +20 cm (NASA), -25 cm (CICE-ini) and -30 cm (CICE-free). North of Ellesmere Island, CICE-ini indicates positive thickness anomalies (up to +50 cm), whereas all 3 CS2 products show negative thickness anomalies (up to -80 cm). In this region, the CICE-free simulation also shows mostly negative thickness anomalies (-20 to -80 cm), with a small positive area (up to +25 cm).*

2. Line 297-299 I'm not convinced that the snow loading in CS2 has caused this difference in April 2017 north of the Canadian Archipelago and Greenland. If I recall, the Beaufort High collapsed in the winter of 2017 and this reduced convergence against the northern Canadian Archipelago and Greenland which appears to the case in Figure 8. The latter seems more likely the cause of why the thickness anomaly in this region was not preserved as atmospheric circulation prevent dynamic ice growth. This seems to be captured across all CS2 products but not CICE-ini. This needs revision. See major comment.

   **We agree. See our responses to your major comment above, and see the revisions made between lines 316 to 343 pasted below.**

   *On the other hand, thickness is also strongly influenced by dynamics, such as convergence against the CAA and Greenland which leads to thicker ice in this region [Kwok et al., 2015]. During winter 2017 however, the Beaufort High largely collapsed, reducing convergence against the northern CAA and Greenland [**Figure 8**]. One advantage of using CICE, is that we can more readily diagnose thermodynamic vs. dynamical contributions to the observed thickness anomalies. For the region directly north of Ellesmere Island, both the CICE-ini and CICE-free simulations support reduced sea ice convergence, leading to thinner ice from dynamical contributions. At the same time, this region also exhibited reduced thermodynamic ice growth in both CICE simulations. One would expect thermodynamic ice growth to be reduced in regions of enhanced snow depth and thicker November ice. Positive snow depth anomalies extended from this region through the northern Beaufort Sea, in agreement with extended regions reductions in thermodynamic ice growth in both CICE-free and CICE-ini. At the same time, regions of positive 2016 November thickness anomalies are also associated with regions of reduced CICE thermodynamic ice growth.*

3. Line 413-415 The snow is important but ice thickness is strongly influence by dynamics (i.e. convergence against the Canadian Archipelago and Greenland) and this needs to be mentioned in the discussion as well. See Kwok, 2015, GRL. Furthermore, MYI is the dominant ice type north of Greenland and the Canadian Archipelago which has consistently been preserved despite the shift from MYI to FYI elsewhere. This suggests the snow depth here should be similar to the W99 which was found reported by Haas et al., 2017, GRL hence CS2 estimates here should be reliable and lends further support to reduced ice convergence was more influential on thinning. See major comment.
   **See our responses to your major comment above.**

4. Table 1 What is the source of the data in this table? The passive microwave algorithm from Markus et al., 2009, JGR?
   **Yes, from Markus et al. 2009 and from Stroeve et al., 2014. References were mentioned in the body text, but now also added to the Table caption.**

5. References: Haas, C., Beckers, J., King, J., Silis, A., Stroeve, J., Wilkinson, J., Notenboom, B., Schweiger, A., & Hendricks, S. (2017). Ice and snow thickness variability and change in the high Arctic Ocean observed by in situ measurements. Geophysical Research Letters, 44, 10,462–10,469. https://doi.org/10.1002/2017GL075434 Kwok, R. (2015), Sea ice convergence along the Arctic coasts of Greenland and the Canadian Arctic Archipelago: Variability and extremes (1992–2014), Geophys. Res. Lett., 42, 7598–7605, doi:10.1002/2015GL065462.
   **Thank you, these have been added.**

---

## Author Comment (AC2) · 27 Feb 2018

We thank the reviewer for their thoughtful comments and our responses are shown in red below.

Summary

This paper uses model simulations from the Los Alamos sea-ice model (CICE) and CryoSat-2 thickness estimates from three different data providers to investigate the impact of the 2016/2017 anomalously warm winter on Arctic sea ice thickness. The authors consider free CICE simulations as well as CICE simulations initialized with CryoSat. Coinciding with the least amount of freezing degree days north of 70N since 1979, the authors find that CICE simulations in April 2017 show the thinnest ice cover in the Arctic Basin over the CryoSat-2 data period. However, this finding is not entirely supported by the satellite retrievals. CICE simulations are also used to investigate the processes leading to ice thickness anomalies, separating dynamic and thermodynamic contributions. It is concluded that free CICE simulations from 1985 to 2017 reveal that the correlation between winter ice growth and November ice thickness is stronger than between growth and FDDs, although this correlations has become weaker since 2012, and delayed freeze up due to warmer winter temperatures play a bigger role.

General comments:

The impact of warmer winter seasons on the Arctic ice cover is of high interest for the sea ice and climate science community. In addition, the comparison between sea ice thickness retrievals from different providers adds some valuable information here. The manuscript itself is well written, but there are lots of information in the figures and tables which are not easy to capture. For example, color bars in Figure 4 show different scales, which is a bit confusing. Also the quality of the figures in general can be improved. See more detailed comments below.

Apart from that, my major concern is that it is **not really well explained how reliable the model simulations are**, both CICE free and CICE initialized with CryoSat. Although the mean monthly values seem to fit quite well to the satellite observations, considering Figure 3 and Figure 5, **regional anomalies disagree quite significant in several cases**. For example, the significant positive thickness anomaly north of the Canadian Archipelago in April 2014 and 2015 is rather weak in the model simulations. I don't think that this is due to the usage of a snow climatology in the satellite retrievals, since this area is mostly covered by multiyear sea ice. I also wonder why this strong positive anomaly is not present at least in the CICE simulations initialized with CryoSat. Based on these concerns, I also wonder **how reliable the findings and conclusions regarding the results presented in Figure 9** are. Could you include the satellite observations here as well? Also difference maps and scatter plots between simulated ice thicknesses and CryoSat ice thicknesses would be interesting and could potentially help to support the conclusions and show more explicit the limitations of the model simulations. For example, how meaningful are the correlations given in the maps of Figure 9 if the model is limited in reproducing regional anomalies as described above?

Local and to a lesser extent regional results from our model simulations are affected by a variety of uncertainties, including slightly shifted location of moving cyclones can result in wrong pattern of ice drift and ice divergence, and reanalysis precipitation likely has biases as well. Thus, we do not believe, nor do we state that all the small regional features shown in

the maps in Fig. 4 to 6 are realistic. At this scale we are only confident for regions where CryoSat-2 products and CICE simulations agree (see original paragraphs lines 263-285). In Figure 9, however, we are looking at an Arctic Basin wide mean. For the Arctic Basin wide mean, thermodynamic processes are dominating over the dynamic processes (see Table 3) and the thermodynamic winter ice growth has been tuned successfully to agree with the Cryosat winter ice growth. Thus, our results on this scale are reliable as further demonstrated by Fig. 1b. There are no satellite observations of ice thickness available which cover a period of more than 30 years and thus, it would not be correct to use those for Figure 9 as the time-period is simply too short for meaningful correlations. We have added a comment on lines 268-270 to highlight that fact up front (*While we discuss some of the regional differences below, we are most confident in the model simulations on the Arctic Basin-wide scale over which CICE has been tuned to agree with CS2 winter ice growth*.).

In response to the comments on the plots and color bars, we have made improvements that hopefully satisfy the reviewers concerns.

Detailed comments:
P3 L109: The CPOM product is derived using a 70 % threshold, not 50 % as stated in this paper (and in Laxon et al. (2013) because of a typo). There is an erratum for Laxon et al. (2013) where a 70% threshold is reported.
Thank you for pointing this out, it has now been corrected.

P3 L124: Category 1 ranges up to 0.6 m. But when you discard any measurements below 0.5 m, then you this category only covers a very narrow range of thickness. Isn't that a limitation for the initialization of the model then?
We discard grid point with a mean thickness below 0.5m, but otherwise we include all individual measurements. We state that "Grid points with less than 100 individual measurements and a mean SIT < 0.5 m are not included." But have now added the extra statement to avoid confusion: "Otherwise, all individual observations are included"

P3 L138: CICE simulations - What are the grid cell ice thicknesses in the CICE simulations representing? The mean thickness of the ice covered area or the mean thickness of the entire area including open water? This information should be given in this section, because it is crucial when comparing it with the satellite data.
This is a good point. We have now added at the end of this section the statement: "For comparison with CS2 we present the mean thickness of the ice covered area. In winter the sea ice concentration in the model is generally between 0.98 and 0.995% apart from locations close to the ice edge".

Figure 1 c): Information about the red and the yellow areas is missing.
Corrected.

Figure 2, L677: I cannot see any light gray areas. The legend in Fig 2c is very small.
We have increased the size of the legend. We removed the statement about the light gray areas as they are actually shown in white in Figure 2d. Here is the new Figure 2c.

[Figure]

Figures 3, 5, 6, 11: The labels of the color tables are too small. Since all maps of each figure correspond on the same thickness range, I suggest to use just one color bar and make it bigger.
We have removed the individual color bars and now just use one larger horizontal color bar.

Figure 4: It is a bit confusing that you use different thickness ranges for the CICE anomaly contributions from thermodynamics and dynamics (+/- 0.4), while for the other maps, you use +/- 0.8. I suggest to use a uniform range, e.g. +/- 0.8. This would make a comparison with the other maps easier.
We agree and made the suggested change.

Second, I wonder how to interpret the thermodynamic and dynamic contributions. For example, there is a positive CICE anomaly north of the archipelago (middle left), while both the thermodynamic (middle center) and dynamic (middle right) contributions show negative anomalies. How is this explained?

Well, in your example a very strong positive CICE anomaly in Nov 2016 (Fig. 3) has been reduced by thermodynamic and dynamic processes (positive anomalies) to result in a weaker, but still positive anomaly in April 2017. Thus, the initial conditions in November are responsible. Thermodynamic contribution consists of local ice growth/melt and dynamic contribution of advection and ridging processes during the period November to the following April.

Moreover, there is a typo in the caption (L692). I suppose contribution of dynamics is shown in the "right" column.

Yes, thank you.

---

## Author Response (AR2)

Response to reviewers comments

Reviewer 1

General Comments

The authors have nicely responded to my concerns on ice dynamics but I still have a concern about the snow thickness role in 2017. While I agree that snowfall will vary year to year, it is puzzling the authors do not even make reference to actual in situ measurements of snow thickness in their region of interest. In Figure 7 (2017 panel), I see no reason why the CICE snow anomalies should not be compared to actual in situ measurements made during that time by Haas et al., (2017). Those measurements will provide some regional insight into how representative CICE April 2017 snow anomalies actually are and in turn give stronger (or weaker) support for the role of snow thickness in April 2017 ice thickness anomalies. Reliance on reanalysis data should not take precedence over in situ observations if they are available.

We appreciate the reviewers comments, yet doing the comparison within the paper is somewhat problematic as it would invite a systematic comparison of CICE snow depth with all available in situ data, something outside the scope of this paper. In addition, we are working with monthly CICE output whereas the in situ data was collected over a week period, and as we know, a single in situ observation is unlikely to represent the larger CICE grid cell. Nevertheless, we have looked at the mean snow depth for each site collected (i.e. average of all transects at each site) during the Lincoln Sea field campaign and compared that with the mean monthly CICE data for April 2017. We find that in general the monthly mean value of April snow depth from CICE is less than the snow depths measured for days during April in the Lincoln Sea, though the standard deviations are high in the in situ data (about half the mean). We added a statement to that effect in the text Lines . "While it is difficult to ascertain the accuracy of the CICE snow depths, comparison with snow depths collected by *Haas et al.* (2017) in the Lincoln Sea showed CICE snow depths were underestimated on average by 10cm (not shown). Thus, the overall impact on overall ice thickness may be less than computed here.

[Figure]

[Figure]

Line 62-62
Ricker et al. [2017a] is initially referenced and then Ricker et al. [2017b]. Suggest putting them both at the beginning or end.
The reason for putting the first one at the start and the second at the end is that the second paper pertains to blending the CS2 data with SMOS, whereas the first reference pertains to the study where they used the data to examine thickness changes. Thus, we have left it unchanged, but can change if the editor prefers.
Line 360
Reference to the very recent (and timely) Moore et al. 2018, GRL paper should be made here
about the collapse of the Beaufort High.
Done.
Line 375
CAA not CCA
Done

Reviewer 2
The authors have mostly addressed my concerns. The readability of the figures is improved and
clarifications have been made in the text.
I only have one minor concern left. I am still not fully convinced of how the model simulations
are discussed in the paper. I understand that you only consider the Arctic Basin and that for this
area, the mean monthly ice growth rates match the satellite observations very well. And I think
for most parts of the paper, this works out, when you consider Arctic Basin wide ice growth
rates. But I have the feeling that sometimes the model simulations are considered as the
"reference". To be more specific, for example, after discussing differences in regional patterns
between the products (L301-331), in L332-334: "While the discrepancy in this region is
puzzling, the bias between the CICE-ini simulations and the CS2 products may in part reflect the
use of a snow climatology in the CS2 thickness retrievals". Surely, this can be a reason, but it
could be also due to the model forcing, no? My point is that the presented evaluation of the
model simulations only shows that the mean monthly thickness of an Arctic sub-region matches
the satellite observations. Therefore, I would like to see a clear statement about the
limitations/uncertainties of the model simulations in the paper, when differences as in L301-331
are discussed.
Again, I think the paper is well written and interesting, and deserves publication in TC.
We appreciate the reviewer's comments. We agree with the reviewer that the model simulations
are not the reference and biases between the CICE simulations and CS2 may also be from model
biases. However, snow cover can also be one of the reasons and unfortunately none of the
CryoSat-2 algorithms can currently differentiate between anomalous snow coverage and
anomalous sea-ice thickness. Since the other reviewer also raised the concern about the snow
depth in CICE, we looked at our 2017 snow depths relative to the in situ data collected in the
Lincoln Sea and found they are generally underestimated, yet this is not a very robust assessment
since we looked at monthly CICE data and compared with daily data collected in April.

It is further important to note that it is impossible to derive precise error estimates for simulated ice thickness at each grid cell. We add this sentence on lines 284-286 to address the uncertainties "
[revised manuscript text omitted]

| 2011 | 239 (-4) | 237 (+7) | 227 (-8) | 242 (-4) | 241 (+1) | 115 (+3) | 104 (+1) | -18 (-3) | -20 (-3) |
| 2012 | 235 (-8) | 219 (-11) | 218 (-17) | 247 (+1) | 233 (-7) | 115 (+3) | 110 (+7) | -9 (+6) | -12 (+5) |
| 2013 | 230 (-13) | 208 (-22) | 210 (-25) | 234 (-12) | 237 (-3) | 136 (+24) | 117 (+14) | -16 (+1) | -19 (-2) |
| 2014 | 261 (+18) | 250 (+20) | 254 (+19) | 251 (+5) | 249 (+9) | 102 (-10) | 94 (-9) | -12 (+3) | -17 (+0) |
| 2015 | 264 (+21) | 252 (+22) | 254 (+19) | 264 (+18) | 255 (+11) | 108 (-4) | 103 (-0) | -18 (-3) | -22 (-5) |
| 2016 | 239 (-4) | 227 (-3) | 228 (-7) | 254 (+8) | 241 (+1) | 107 (-5) | 101 (-2) | -15 (-0) | -17 (+0) |
| 2017 | 230 (-13) | 218 (-12) | 238 (+3) | 233 (-13) | 227 (-13) | 99 (-13) | 92 (-11) | -14 (+1) | -13 (+4) |